# Ferroptosis: A Novel Type of Cell Death in Male Reproduction

**DOI:** 10.3390/genes14010043

**Published:** 2022-12-23

**Authors:** Yanjing Su, Zelan Liu, Keyu Xie, Yingxin Ren, Chunyun Li, Wei Chen

**Affiliations:** 1The Key Laboratory of Model Animals and Stem Cell Biology in Hunan Province, Hunan Normal University School of Medicine, Changsha 410013, China; 2Department of Clinical Medicine, Hunan Normal University School of Medicine, Changsha 410013, China; 3Department of Nursing, Hunan Normal University School of Medicine, Changsha 410013, China

**Keywords:** ferroptosis, iron metabolism, GPX4, lipid peroxidation, male reproduction, male infertility

## Abstract

Ferroptosis, an iron-dependent type of regulated cell death, is triggered by the accumulation of lethal lipid peroxides. Due to its potential in exploring disease progression and highly targeted therapies, it is still a widely discussed topic nowadays. In recent studies, it was found that ferroptosis was induced when testicular tissue was exposed to some high-risk factors, such as cadmium (Cd), busulfan, and smoking accompanied by a variety of reproductive damage characteristics, including changes in the specific morphology and ferroptosis-related features. In this literature-based review, we summarize the related mechanisms of ferroptosis and elaborate upon its relationship network in the male reproductive system in terms of three significant events: the abnormal iron metabolism, dysregulation of the Cyst(e)ine/GSH/GPX4 axis, and lipid peroxidation. It is meaningful to deeply explore the relationship between ferroptosis and the male reproductive system, which may provide suggestions regarding pristine therapeutic targets and novel drugs.

## 1. Introduction

It is known that cell death is classified as accidental cell death (ACD) or regulated cell death (RCD) [1]. Distinguished from ACD caused by an exposure to severe physical, chemical, or mechanical damage, the latter is capable of being regulated pharmacologically or genetically. RCD was mainly studied in three categories—apoptosis, autophagy, and necrosis—before ferroptosis was officially proposed and became a research hotspot. Ferroptosis, a nonapoptotic type of RCD closely related to the iron metabolism and lipid metabolism, is implicated in infection, inflammation, and immunity [2], especially in the cancer pathogenic mechanism [3]. Notably, lipid peroxidation is the execution step of ferroptosis, and it can be induced by the overload of iron, barriers of glutathione peroxidase 4 (GPX4) utilizing glutathione (GSH), and the excessive accumulation of lipid reactive oxygen species (ROS) [4]. Overall, ferroptosis is a complex and general process arising from various mechanisms, and its potential in exploring disease progression and highly targeted therapies has made it a research hotspot in recent years.

With the ever-expanding field of ferroptosis, it has also been proven that ferroptosis also occurs in the reproductive system [5,6,7]. Different factors induce ferroptosis, resulting in the morphological structure or functional dysfunction of reproductive cells, which is closely related to the occurrence of associated diseases, such as male azoospermia and oligospermia [5,8]. Although many experimental studies had been reported, an updated review to summarize the relationship between the male reproductive disorders and ferroptosis is largely missing. We herein researched the related literature in the PubMed database (Appendix A) and reviewed the correlated processes of ferroptosis in male reproduction, especially the abnormal iron metabolism, regulation of GPX4, and lipid peroxidation; we also detailed the challenges and presented several new ideas in the male reproductive field, which could provide a new target for the treatment of male infertility.

## 2. The Discovery of Ferroptosis

In 2003, Professor Stockwell and his research team from Columbia University initially discovered a novel type of cell death induced by erastin. This offered an entry point into a new domain, because they found that this form of cell death was not accompanied by the activation of traditional caspase or DNA fragmentation, which are characteristics of apoptosis [9]. However, the internal mechanism was unclear. The research group carried out a series of experiments on cells in the presence of oncogenic RAS with an erastin treatment. It was not until 2007 that the mechanism of erastin-induced death was revealed. They concluded that erastin functioned through mitochondrial voltage-dependent anion channels (VDACs) and it was related to the extracellular RAS–RAF–MEK–ERK pathway which belongs to the mitogen-activated protein kinase (MAPK) signaling pathway [10]. At the beginning of 2008, the group identified two distinctive compounds, named RAS-selective lethal 3 (RSL3) and RAS-selective lethal 5 (RSL5), which displayed a higher lethality in two engineered BJ-fibroblast-derived cell lines harboring RAS ^V12^. Interestingly, RSL3 and RSL5 have an erastin-like function, which induces a particular process of cell death that is accompanied by increased levels of intracellular ROS and a higher abundance of the iron content. Differently, RSL5 acts in a VDAC-dependent manner while RSL3 acts in a VDAC-independent manner [11]. It is appropriate to mention that RSL3 is still widely used as a classic ferroptosis agonist by repressing GPX4, while RSL5 is rarely used. The year 2012 represented a milestone for ferroptosis. In this year, the unique iron-dependent form of cell death caused by erastin was officially named “ferroptosis” by Professor Stockwell and his team [12]. Although more than ten years have passed, ferroptosis is still a relevant topic due to its key role in the physiological and pathological processes [4]. As ongoing experiments are being widely conducted, our appreciation of the depth of the functions and mechanisms of ferroptosis continues to be updated.

## 3. Iron Metabolism and Ferroptosis in the Male Reproductive System

The cellular iron metabolism is essential for maintaining life activities in normal biological processes, including the generation of cellular energy, oxygen transport, heme synthesis, DNA synthesis and repair [13], etc. Notably, studies show that the iron metabolism is also quite important for the reproductive system and is intimately related to spermatogenesis [14]. The relationship between the abnormal iron metabolism and ferroptosis in male reproductive disorders is gaining increasing attention.

### 3.1. Cellular Iron Metabolism

The mammalian iron metabolism could be divided into four sections: absorption, storage, utilization, and efflux. The main mechanism for the intake of extracellular iron is a transferrin-bound iron (TBI) import, which relies on the iron transporter transferrin (Tf) and the transmembrane glycoprotein Tf receptors (TfR). The other mechanism is the non-transferrin-bound iron (NTBI) input, which depends on the cellular membrane protein represented by SLC39A14(ZIP14) [15]. First, each Tf is combined with two extracellular ferric ions (Fe^3+^), which subsequently bind to the TfR in TBI. Then, the Tf-TfR complex is internalized into the endosome. As the environment in the endosome is acidic, ferric ions are released from the complex, while Tf and TfR will be circled to the cell surface for the next round of iron absorption [16]. Next, the ferric ions are rapidly reduced to ferrous ions (Fe^2+^) by STEAP3 metalloreductase. Following this, Fe^2+^ would be exported to the cytoplasm from the endosome by a divalent metal transporter 1 (DMT1) and forms the labile iron pool (LIP). Iron in the LIP is loosely bound to negatively charged small molecules and proteins [17], so its catalytic activity is difficult to control. Notably, LIP is the transportation hub that links the iron metabolism mentioned above. The iron could exit the LIP in one of the four main pathways: (i) the majority of iron is transported to ferritin, which is a ubiquitous iron-storage protein in the cytoplasm and can combine up to 4500 iron atoms [18]. (ii) Iron can also be utilized directly in the cytoplasm as a cofactor of cytoplasmic proteins. (iii) Iron could be trafficked to the mitochondria via SLC25A37 and SLC25A28 to participate in heme and iron-sulfur (Fe-S) cluster synthesis [19] or to be stored in mitochondrial ferritin (MtFt) [20]. However, heme could be decomposed into Fe^2+^ under the catalysis of heme oxygenase 1(HO-1), which increases the LIP in turn [21,22]. It is worth noting that mitochondrial iron is crucial for spermatogenesis. Research revealed that the number and vitality of sperm were reduced in mitoferrin-2 knockout mice [23]. Furthermore, the down-regulation of the mitoferrin gene resulted in germ cell developmental defects in Drosophila, some of whose mitochondrial iron metabolism genes are similar to mammals in their expression patterns [24]. (iv) Excess cellular iron in the LIP would be released outside the cell by ferroportin (FPN), which is the only known cellular iron exporter (Figure 1).

In addition to the general iron metabolism, there is a largely unique autonomous iron cycle in spermatogenesis (Figure 1). Iron is taken up by spermatogonia and preleptotene spermatocytes via TBI and continuously transferred to round spermatids with their self-renewal. In particular, a proportion of iron is released by round spermatids and cycled to the apical compartment of the Sertoli cells, with the rest carried away by elongated spermatids. Sertoli cells store the acquired iron in ferritin, which would be secreted and returned to primary spermatocytes. Although the elongated spermatids would take a part of the iron away from seminiferous epithelia, spermatogonia and preleptotene spermatocytes could absorb iron to compensate for the loss via the TBI mentioned above. This internal iron cycle protects the developing germ cells from peripheral iron fluctuations, thus ensuring the stability of spermatogenesis [25].

### 3.2. Abnormal Iron Metabolism Leads to Dysfunction of Male Reproductive System

The abnormal iron metabolism, either an iron excess or iron deficiency, would affect spermatogenesis and cause a reproductive system dysfunction. Here, we mainly discuss the damage caused by an iron overload in the male reproductive system. The principal causes of an iron overload in testicular cells are the degradation of the reservoir or obstruction of the outlet. Once ferritin is degraded through autophagy under the regulation of the autophagic receptor nuclear receptor coactivator 4 (NCOA4), the large number of iron atoms stored in it would be released to the LIP [26]. The degradation of FPN, whether mediated by acknowledged hepcidin [27] or newly discovered ubiquitin ligase RNF217 [28], hinders the iron output and increases the internal iron content as well. As mentioned above, the LIPs catalytic activity is unrestricted when iron-overloaded; thus, ferroptosis in testicular cells can be triggered in the following two ways. On the one hand, electrons can be transferred to hydrogen peroxide via the Fenton reaction, Fe^2+^ + H_2_O_2_→Fe^3+^ + OH^−^ + •OH, forming the extremely damaging hydroxyl radical [29]. On the other hand, iron-dependent oxidases, including lipoxygenases, xanthine oxidases, NADPH oxidase, or catalase [30] could generate soluble and lipid ROS to promote ferroptosis (Figure 1).

Ferroptosis is a newly discovered cause of clinical diseases of the male reproductive system. A study showed that ferroptosis was involved in the occurrence of asthenozoospermia and led to an impaired sperm function. The iron levels in the semen of asthenozoospermic patients were significantly higher than those of the normal group, accompanied by an increase in ROS [8]. Furthermore, it is noteworthy that smokers have higher levels of ferroptosis than non-smokers among infertility patients. Heavy smokers have significantly more iron and ROS in seminal plasma and problems with sperm abnormalities, such as sperm vitality and progressive motility [6].

The down-regulation of FPN is particularly noticeable in animal and cell models, created for studying male reproductive system disorders. After an exposure to low-dose Cd, the expression of FPN was significantly down-regulated, with an elevated total iron and ferrous iron levels in testicular cells. This exposure also resulted in the massive loss and detachment of testicular germ cells, as well as a decreased meiotic index and testicular weight in mice [31]. However, the expression of the input proteins—not only TfR but also ZIP8 and ZIP14—was not altered. Interestingly, ZIP8 and ZIP14 played major roles in the uptake of NTBI, and the iron input in this way increased the potentially harmful LIP in liver cells [15,32]. Furthermore, Zhao et al. found that a busulfan exposure led to ferroptosis in mice testes, which was manifested by the down-regulation of FPN, the accumulation of iron, an increased malondialdehyde (MDA) content, and the morphological changes. They further confirmed that ferroptosis was caused by down-regulated FPN and GPX4. After treatment with deferoxamine (DFO), an iron chelator, the iron content and abnormal changes mentioned above were all reversed [5]. The decreased expression of the FPN protein and mRNA resulted in elevated iron and lipid ROS levels, and ferroptosis was inhibited by either a DFO treatment or FPN overexpression in ischemia–reperfusion-injured TM4 cells, which were the testicular Sertoli cells of normal mice [33]. A significant increase in intracellular iron and ferroptosis also occurred in the TM4 cells exposed to PM2.5, which are fine particles with a diameter of 2.5 μm or less [7]. Based on the existing studies, we speculate that FPN is vulnerable to risk factors, leading to iron accumulation-induced testicular ferroptosis.

Different testicular cells are differentially sensitive to iron accumulation-induced ferroptosis. Round spermatids showed iron-dependent cell death induced by 4-hydroxynonenal (4HNE), while pachytene spermatocytes could not be induced to ferroptosis by 4HNE [34]. In Leydig cells, tetramethyl bisphenol A (TMBPA) significantly increased the testicular iron content, induced ferroptosis, and inhibited the testosterone synthesis during late puberty [35]. An iron/fat-enriched diet led to the degeneration of seminiferous tubules in mice, an increase in vacuoles and space in the basement membrane, and a decrease in the total testosterone levels in the testis. Because testicular iron and hepcidin were up-regulated and the FPN expression was down-regulated, we suspect that ferroptosis also occurred in Leydig cells, although this experiment only demonstrated that they underwent apoptosis after the treatment [36].

Contrary to the above, the decrease in the iron content in the LIP would inhibit ferroptosis [37], but it is still unfavorable to spermatogenesis. A lack of iron could lead to iron deficiency anemia (IDA), which induces an anoxic environment in the testis. An experimental article reported that after the correction of IDA, the patients’ testosterone levels and sperm parameters were significantly improved, especially in those who had subnormal values of these parameters [38]. Interestingly, the latest research shows that an iron supplementation could abrogate the testis dysfunction due to an iron deficiency through the regulation of the testicular antioxidant capacity [39].

In summary, we conclude that the stabilization of the intracellular iron balance is essential to maintain the normal process of spermatogenesis. An iron overload leads to ferroptosis in germ cells, and inhibiting it, such as by using iron chelation or up-regulating the expression of FPN, is likely to provide an effective approach to reversing this type of death.

## 4. The Cyst(e)ine/GSH/GPX4 Axis

Except for the iron overload mentioned above, the inactivation of the cellular antioxidant system is also a primary cause of ferroptosis. Based on the current findings, there are three main axes in the cellular antioxidant system: the Cyst(e)ine/GSH/GPX4 axis, the GCH1/BH4/DHFR axis, and the FSP1/CoQ10 axis [40]. Among them, the Cyst(e)ine /GSH/GPX4 axis is the most well-studied one. In this section, we introduce the Cyst(e)ine/GSH/GPX4 axis and summarize the present understanding of the main antioxidant system in the male reproductive system.

### 4.1. GSH Metabolism

The biosynthesis of reduced GSH cannot be ignored when referring to the Cyst(e)ine/GSH/GPX4 axis. Cys2, Gly, and Glu, the main raw materials of GSH biosynthesis, could be transported into cells, respectively, by systemXc-, AT1, and AT2, which are located on the cell membrane [41]. Once inside the cell, Cys2 is reduced to Cys by thioredoxin reductase 1 (TrxR1) immediately [42]. Then, Cys and Glu generate γGlu-Cys under the assistance of γ-Glu-Cys ligase (GCL), which is the rate-limiting enzyme of GSH synthesis. Consequently, Gly is added to the C-terminal of the γGlu-Cys catalyzed by glutathione synthetase (GSS), then forming GSH [43]. In biological reactions, the G-SH can reduce selenic acid (-SeOH) to -SeH, which is the main active group of GPX4, and further release GS-SG to prevent GPX4 from being inactivated. Meanwhile, the phospholipid hydroperoxides (PLOOH) could oxidize -SeH to -SeOH and be returned to harmless hydroxyl derivatives [44]. Intracellular GSH or GSSG would be extruded outside via a multidrug resistance-associated protein (MRP/ABCC) [12]. They are hydrolyzed to Cys2, Gly, and Glu by γ-glutamyl transpeptidase (GGT) and dipeptidase (DP), which face the outside of the cell membrane [43]. In this way, three raw materials are recycled for a subsequent GSH biosynthesis (Figure 2). The whole process is as follows:2G-SH + GPX4-Se-OH→GS-SG + GPX4-SeH + H_2_O,(1)
PLOOH + GPX4-SeH→PLOH+GPX4-Se-OH.(2)

### 4.2. SystemXc- and GSH Associated with Ferroptosis in the Male Productive System

SystemXc-, as a Cys2/Glu antiporter in the upstream of this axis, is composed of catalytic subunit xCT/Solute Carrier Family 7 Member 11 (SLC7A11) and regulatory subunit 4F2 (4F2hc)/Solute Carrier Family 3 Member 2 (SLC3A2), of which SLC7A11 is the main active function subunit [44]. Normally, intracellular Cys is maintained at a low concentration, so it is critical for Cys to be taken in by systemXc- for the GSH synthesis [43]. In this case, suppressing the expression of systemXc- could reduce the intracellular Cys content, which directly leads to a reduction in the GSH production. Then, the activity of GPX4 in the cells would be affected by the GSH fluctuation directly, and this finally leads to the occurrence of ferroptosis [45].

Studies have shown that SLC7A11 is highly expressed in Sertoli cells [46,47] and the *Slc7a11* knockout mice presented a high proportion of immature sperm in the cauda epididymis due to an insufficient cysteine supply [48]. Moreover, ferroptosis induced by TMBPA in rat Leydig cells occurred not only due to the above-mentioned iron overload, but also the down-regulation of the expression of SLC7A11 and GPX4 [35]. Furthermore, the number of frozen–thawed stallion sperm bound per oocyte tended to increase after the post-thaw incubation with Cys2 compared to the treatment with Sulfasalazine (SS), a specific inhibitor of the SLC7A11 antiporter [49]. These examples strongly demonstrate that the susceptibility to ferroptosis would increase when inhibiting the expression of systemXc- in testicular cells.

As for downstream GSH, it is representative of the non-enzymatic antioxidant defense systems in the body and is normally recycled to sustain its antioxidant capacity. Inside the testes of rats and mice, high levels of GSH have been found, underscoring the importance of GSH in the reproductive system [50]. A study showed that GSH could improve the spermatozoa quality and testicular histomorphology in diabetic mice [51]. However, as GSH is unable to be taken into cells extracellularly, its concentration is controlled exclusively by intracellular synthesis [52], and the effect of its oral treatment is proven to be limited [53]. Until now, there are few reports on GSH in the treatment of male fertility, and it is still worth exploring how exogenous GSH effectively exerts its lipid peroxidation compacity to combat ferroptosis.

### 4.3. Effects of GPX4 Associated with Ferroptosis in Testicular Cells

Although GPX4 requires some assistance from its upstream GSH to maintain its reducing state, it is GPX4 itself that directly exerts its anti-lipid peroxidation activity. Consequently, a great deal of attention has been paid to GPX4, which is widely recognized as the guardian of the ferroptosis kingdom [54].

GPX4 plays an irreplaceable role in maintaining the normal spermatogenesis process. In earlier stages of spermatogenesis, GPX4 could protect against oxidative stress-induced injury and act as a provider of secure microenvironments. In later stages, in addition to its antioxidant effect, the immunochemical analyses of mouse seminiferous tubuli illustrate that late spermatids and mature sperm had a significant increase in GPX4 [55], which becomes a structural component of mitochondrial capsules in the midpiece to maintain the stability and motility of sperm [56]. It is hypothesized that the significant decrease in the sperm density and viability resulting from the down-regulation of the GPX4 expression is due to two factors: its importance for the structure of the sperm dynamic system and the risk of a cellular exposure to ferroptosis damage during spermatogenesis as a result of its antioxidant failure. In addition, GPX4 is a selenoprotein distinctly expressed in the testes, and it was demonstrated that a selenium supplementation in selenium-deficient germ cells significantly increased the GPX4 expression, further increasing the proliferation rate [57]. Thus, a selenium supplementation may reverse the GPX4 deficiency.

There is a certain organic correlation among ferroptosis, the GPX4 activity, and male infertility. It was found that 30% of oligoasthenozoospermia patients had a decreased expression of GPX4 in their sperm [58]. Coincidentally, it was also found that the asthenozoospermia group showed lower levels of SLC7A11 and GPX4 than the control group. Moreover, the mRNA expression of GPX4 was negatively related to the iron level mentioned above, which collectively promoted ferroptosis, contributing to the negative impact of the progressive and total motility of sperm [8]. It is not surprising that spermatocyte-specific *Gpx4* knock-out mice became sterile, although they could mate normally [59]. In mice testes treated with different drugs, it was further confirmed that an abnormal GPX4 expression led to ferroptosis. A treatment with bisphenol A would cause oxidative damage and decrease the sperm quality in the testis, and it was confirmed to affect the expression of the ferroptosis-related genes, such as *Gpx4*, acyl-CoA synthetase 4 (*Acsl4*), and so on [60]. In mice with oligospermia induced by busulfan, researchers found that ferroptosis could be partly induced by down-regulating the nuclear factor-E2-related factor 2 (*Nrf2*) [5]. Significantly, it was confirmed that *Nrf2* could regulate GPX4 and systemXc-, whose down-regulation would lead to ferroptosis [61]. Therefore, we infer that the inhibition of the Nrf2-GPX4 signaling pathway might be a potential strategy for the treatment of male infertility.

In general, the interference of any of the Cyst(e)ine/GSH/GPX4 axis members will increase the sensitivity of cells to ferroptosis. Consequently, when considering the inhibition of ferroptosis as a treatment for male infertility, we should attach great importance to the Cyst(e)ine/GSH/GPX4 axis.

## 5. Lipid Peroxidation

Lipid peroxidation is the most pivotal step in executing ferroptosis, which is mainly due to the disorder of the generation and degradation of ROS. The accumulation of ROS causes damage to cell components such as the lipids, proteins, and DNA [12], and its attack towards membranes with high levels of polyunsaturated fatty acid (PUFA) will be lethal [62]. As a free-radical-driven reaction, lipid peroxidation is initiated by a lipid reaction with ROS. Phospholipid hydroperoxides are a form of ROS mediating the lipid peroxidation. The process of PUFAs synthesizing PLOOH through the lipid peroxidation chain could be a bridge to connect metabolic disorders and ferroptosis.

### 5.1. The Common Executor of Ferroptosis—Lipid Peroxidation

When the capacity of intracellular antioxidant systems is not sufficient to cope with excessive lipid ROS, lipid peroxidation against phospholipid bilayers containing PUFA will be provoked. This can occur by both non-enzymatic processes driven by the Fenton reaction and enzymatic processes catalyzed by lipoxygenase (LOX) [63].

Typically, non-enzymatic lipid autoxidation in membranes can be approximately divided into three phases [64] (Figure 2), as follows. (i) Initiation: non-radical phospholipid (PL) is oxidized into a phospholipid carbon-centered radical (PL•) by losing a hydrogen atom. (ii) Propagation: during the process of Fenton chemistry, PL• readily reacts with molecular oxygen to form a peroxyl radical (PLOO) [65], which abstracts hydrogen from allylic carbon and initiates a peroxidative chain reaction. Therefore, PLOOH and new PL• are generated and transmitted. (iii) Termination: a radical–radical interaction inhibits the chain reaction. Moreover, lipophilic antioxidants, including vitamin E, liproxstatin-1, CoQ10 [66], and so on, can eliminate radicals and contribute to termination. In another type, namely the enzymatic process, three enzymes are required in the lipid peroxidation of arachidonic acid (AA) and phosphatidylethanolamine (PE): ACSL4, lysophosphatidylcholine acyltransferase 3 (LPCAT3), and LOX. ACSL4 catalyzes the synthesis of AA-CoA and then it is esterified into AA-PE by LPCAT3. AA-PE is ultimately peroxidized by LOX. LOX comprises nonheme iron-containing dioxygenases encoded by the ALOX gene, which are the most important contributors to the catalysis of a PUFA peroxidation. Based on the positional specificity of oxidation, LOX is classified into different types, and 5/12/15-LOX are widely distributed in human tissues among the LOX family. Arachidonic and linoleic acid serve as the substrates for LOX, whose derivatives are of importance to male reproduction [67]. If the occurrence of lipid peroxidation contributes to abnormal fatty acid profiles, it may be responsible for male infertility [68].

Notably, lipid peroxides can be degraded into toxic products and exhibit an additional toxicity; for instance, peroxyl radicals and alkoxyl radicals can be generated and transfer protons from adjacent PUFAs when lipid peroxides come into contact with transition metals such as ferric or cupric ions [69]. Beyond this, the common decomposition products of a lipid peroxidation within sperm, such as 4HNE and MDA, can alter the structure of the DNA and proteins, contributing to cell injury [70,71].

### 5.2. The Role of Lipid Peroxidation in Male Reproduction

Before the concept of ferroptosis was formally proposed in 2012, studies on the mechanisms related to it, such as the ROS’ toxicity and lipid peroxidation damage in male reproductive pathogenesis, had already yielded some achievements. Strikingly, either the specificity of the germ cell’s composition or the uniqueness of its dynamic changes in the developmental process both indicate that the manifestation of ferroptosis in male reproduction may be unusual. Sperm have a high content of polyunsaturated long-chain fatty acids in the cellular membrane [72], which is a different situation from somatic cells, so it is not difficult to conclude that the sperm are more vulnerable to an attack by lipid peroxides.

Numerous studies have documented that a small quantity of ROS can maintain a normal physiological function [73,74,75]; however, ROS in excess of the limit can be highly toxic. Once the lipid structure is altered, it would lead to a decrease in the membrane fluidity in sperm, thus negatively affecting their motility and the ability to fuse with membranes in fertilization [76,77]. Moreover, sperm have a lack of related cytoplasmic scavenging enzymes compared with other somatic cells [78], thus, when it comes to excessive ROS, they are unable to repair the damage. Nevertheless, abundant antioxidants, such as superoxide dismutase (SOD), catalase, and GPX4, have been proven to exist in seminal plasma, which can compensate for their deficiencies [79].

The susceptibility of different cells in the testis to ferroptosis during spermatogenesis can be roughly judged based on the ROS content and the cell membrane’s sensitivity. Studies demonstrated that the production of ROS was the highest in immature sperm containing cytoplasmic retention and with an abnormal head morphology, and it decreased during the sperm’s maturation until it reached a minimum in mature sperm [80]. Therefore, compared to mature sperm, immature sperm are more susceptible to damage by oxidative stress in this respect, and the probability of ferroptosis may increase. Beyond this, other research also shows the existence of the incorporation of PUFA into the round spermatid membrane during the development of rat pachytene spermatocytes to round spermatids [81]. Furthermore, ALOX15 [82] and ACSL4 [34] also undergo a stage-specific up-regulation in round spermatids, contributing to the sensitivity of the lipid peroxidation, which all points to the specificity of round spermatids for ferroptosis. These findings lead to our hypothesis that the spermatogenic stage from spermatids to mature sperm in seminiferous tubules is more vulnerable to the threat of ferroptosis and may be an unexplained cause of male infertility.

It has been determined that the ferroptosis occurring in the male reproductive system is always accompanied by elevated lipid ROS levels and toxic decomposition products, such as ferroptosis induced by busulfan [5], low-dose cadmium [31], arsenite [83], and so on.

Although ferroptosis is not exactly the same as oxidative stress resulting from the toxicity of ROS, the former usually involves the oxidation of specific lipids, and the threat of ferroptosis to male reproductive health could be resolved by addressing the underlying problem: the breakdown of plasma membranes caused by lipid peroxidation. Thus, potential drugs that are capable of preventing the production of ROS directly or inactivating the ROS to disrupt the executive process may possess a therapeutic value.

### 5.3. Prospective Substances Related to Lipid Peroxidation

Vitamin E, vitamin C, vitamin B9, carotenoids, carnitines, and Q10 have been proven to be able to abort the lipid peroxidation chain reaction, and these common antioxidants play different roles in the male reproductive system to inhibit ferroptosis in germ cells [84]. Beyond this, the intake of zinc may typically function in fighting ferroptosis with its antioxidant properties. It is a main cofactor of SOD and can reduce the production of ROS by displacing iron or copper from membrane binding sites; these are redox-active substances [85] and their deficiency impedes spermatogenesis, which has a negative effect on the serum testosterone concentration [86].

MDA, one of the by-products of the lipid decomposition, mentioned in the preceding paragraph, is often used as a monitoring indicator of ferroptosis and is also used to analyze the degree of peroxidative damage sustained by sperm [87].

## 6. The Changes of Male Reproduction Associated with Ferroptosis

Morphological changes, biochemical features, and the regulation of ferroptotic genes induced by risk factors are representative indicators in ferroptosis.

A morphological examination has a strong reference significance for judging the different types of cell death and diagnosing diseases [88]. Unlike traditional cell death types, ferroptosis, as a nonapoptotic form of cell death, has its own distinctive particularities in morphologic aspects [12]. In 2012, Professor Stockwell and his research team observed BJ-fibroblast-derived cell lines treated with erastin through transmission electron microscopy. Interestingly, they discovered swelling cytoplasm and organelles, ruptured plasma membranes, and chromatin condensation [12]. This was the first exhaustive exploration of ferroptosis in morphology. In recent years, researchers have surprisingly found that smaller mitochondria, an increased mitochondrial membrane density, and reduced or absent mitochondrial cristae can be seen in ferroptotic cells, but not in normal cells [88]. It can be seen that mitochondria are one of the most significant ferroptotic indicators due to their remarkable changes. Even so, the morphological changes in ferroptosis are not set in stone. They may manifest differently depending on the different cell types. For example, no nuclear condensation or chromatin margination appeared in cancer cells with an erastin treatment [88].

The volume of the testes, the morphology of the spermatogenic cells, and the quality of sperms are important indicators for evaluating male reproduction. Many studies have indicated that ferroptosis in the male reproductive system often results in corresponding morphological changes. Here, we have summarized the morphological changes in the reproductive system caused by different risk factors, classed as gross changes and microscopic and sub-microscopic morphological changes (Table 1). In the human, a decreased sperm vitality and progressive sperm motility were observed in heavy smokers compared with non-smokers [6]. When mouse testicles were exposed to certain risk factors that have been proved to be related to ferroptosis, such as busulfan and cadmium (Cd), arsenite, and bisphenol A, the sizes or weights of the mouse testicles decreased [5,31,60,83]. In addition, the irregular morphology of the testicular cells and a decreased quantity of spermatids could be observed at the microscopic level when the mouse testicular cells were exposed to different risk factors [5,7,83]. In particular, a broken or lost mitochondrial ridge could be seen in the testes when adult male mice were treated with arsenite via drinking water [83]. Moreover, an increased membrane density was observed in testicular cells after mice were exposed to bisphenol A via intragastric gavage [60]. The mitochondrial morphology is an important indicator for distinguishing ferroptosis from other forms of cell death.

In addition to morphological indicators, certain changes in the biochemical indicators and levels of the expression of ferroptotic genes can indicate the occurrence of ferroptosis [90]. For example, increases in the iron levels and ROS and MDA contents and the decreases in the expression of GPX4 can often be detected [90]. These indicators are strong indicators of the onset of ferroptosis. Additionally, a decrease in SLC7A11 is discovered when ferroptosis induced by Cd, PM2.5, and arsenite occurs in the male reproductive system [7,31,83]. Additionally, the SOD content is found to increase under ferroptosis induced by Cd, arsenite, and bisphenol A [31,60,83]. As shown in Figure 3, the characteristic changes in the biochemical indicators and ferroptotic gene expression levels due to different risk factors are not identical (Figure 3).

## 7. Therapeutic Potential of Ferroptosis-Related Regulator and Perspectives

As mentioned above, recent research has released signals that some high-risk factors in the male reproductive system, such as the impact of smoking, the induction of arsenite, and even the effects of busulfan, are associated with ferroptosis and mediate the corresponding infertility in men [5,6,83]. Strikingly, all these undesirable effects can be alleviated by Ferrostatin-1(Fer-1) or DFO. Not only that, the exposure of mouse testes to di-(2-ethylhexyl) phthalate (DEHP) can elicit a ferroptotic response through the HIF-1α/HO-1 signaling pathway [91]. The pharmacological modulation of ferroptosis, by inhibiting or inducing it, has been shown to have clinical benefits in specific diseases, especially in cancer, an ischemic organ injury, and degenerative disease therapies. Likewise, according to the recent experiments above, we can find that preventing the lipid peroxidation process or depleting LIP, as targets for any treatment intervention point of ferroptosis, may be of a potential value in treating the reproductive toxicity caused by drugs or toxicants. In addition, the inhibition of ferroptosis targeting the GPX4 pathway might have unexpected effects due to its multiple benefits for germ cells.

Based on the existing research, we proposed the potential mechanism of ferroptosis leading to testicular cell damage and eventually causing a male reproduction disorder (Figure 4). Spermatogenesis is a complex process due to its dramatic cell proliferation and differentiation and apoptosis, which makes the exploration of the mechanisms of nonobstructive azoospermia particularly complex. The questions that we raise are as follows: is there an association between some antioxidants that have been shown to have positive effects on spermatogenesis and ferroptosis? Is the difference in the sensitivity to ferroptosis among different cells during spermatogenesis one of the potential tools to cast light on this mechanism?

Relevant research currently being performed on ferroptosis is limited; due to the gaps in the field of male reproduction, it is crucial to conduct more experiments to explore the contribution of ferroptosis to the germ cell. We look forward to further exploring the underlying pathogenesis of male infertility from the perspective of ferroptosis.

## Figures and Tables

**Figure 1 genes-14-00043-f001:**
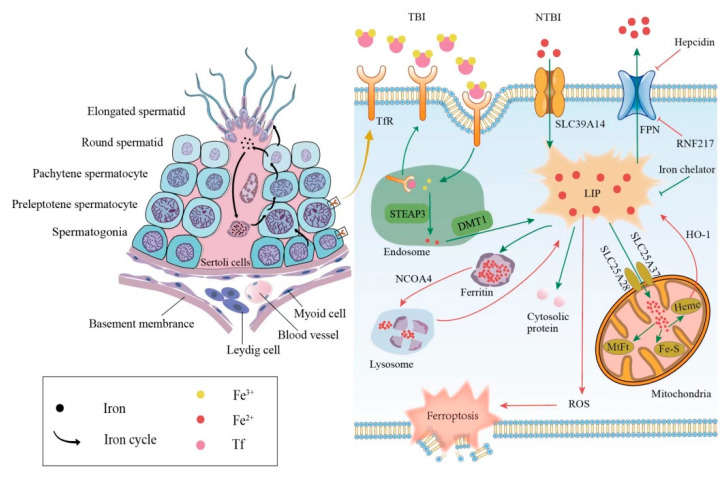
Relationship between iron metabolism and ferroptosis during spermatogenesis. The left side shows the iron circle in the process of normal spermatogenesis. The right side shows cellular iron metabolism and results caused by iron overload. TBI, transferrin-bound iron; NTBI, non-transferrin-bound iron; Tf, transferrin; TfR, transferrin receptor; DMT1, divalent metal transporter 1; FPN, ferroportin; LIP, labile iron pool; MtFt, mitochondrial ferritin; HO-1, heme oxygenase 1; NCOA4, nuclear receptor coactivator 4; ROS, reactive oxygen species.

**Figure 2 genes-14-00043-f002:**
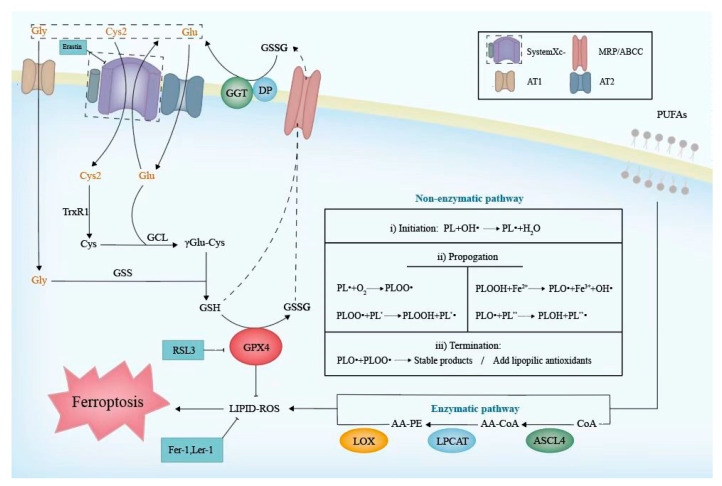
The molecular mechanism of the Cyst(e)ine/GSH/GPX4 axis and lipid peroxidation. The left side of the figure indicates a series of enzymatic reactions for GSH synthesize. As a cofactor for GPX4, GSH could be oxidized to GSSG. Excessive GSH and GSSG could be transported extracellularly. The box on the top right indicates the transporters on the cell membrane. The GPX4 reduces lipid ROS which contributes to the inhibition of lipid peroxidation. The right part depicts the non-enzymatic and enzymatic pathways of lipid peroxidation. The table lists three non-enzymatic phases including initiation, propagation and termination. ACSL, LPCAT3, and LOX are the key enzymes of enzymatic pathways. GSH, glutathione; GCL, γ-Glu-Cys ligase; GSS, glutathione synthetase; MRP/ABCC, multidrug resistance-associated protein; GGT, γ-glutamyl transpeptidase; DP, dipetdase; PUFA, polyunsaturated fatty acid; PLOOH, hospholipid hydroperoxides; PL, phospholipid; PLOO·, peroxyl radical; PLO·, alkoxyl radical; OH·, hydroxide ion; PLOH, phospholipid alcohol; AA, arachidonic acid; PE, phosphatidylethanolamine; ACSL4, acyl-CoA synthetase 4; LPCAT3, lysophosphatidylcholine acyltransferase 3; LOX, lipoxygenase.

**Figure 3 genes-14-00043-f003:**
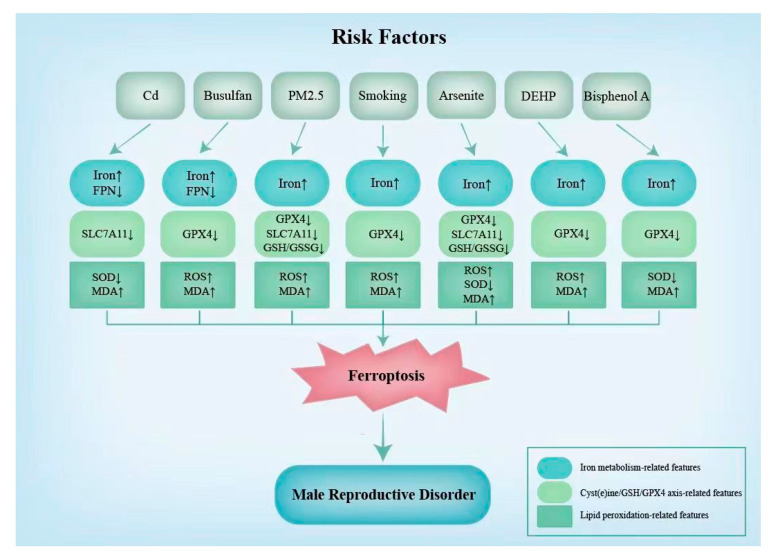
Ferroptosis-related features in male reproduction induced by risk factors. The changes in biochemical indicators and levels of the expression of ferroptotic genes caused by the risk factors associated with ferroptosis are listed in this figure. Cd, cadmium; DEHP, di-(2-ethylhexyl) phthalate; FPN, ferroportin; SLC7A11, catalytic subunit xCT/Solute Carrier Family 7 Member 11; GPX4, glutathione peroxidase 4; GSH, glutathione; SOD, superoxide 2-ethylhexyl smutase; MDA, malondialdehyde; ROS, reactive oxygen species.

**Figure 4 genes-14-00043-f004:**
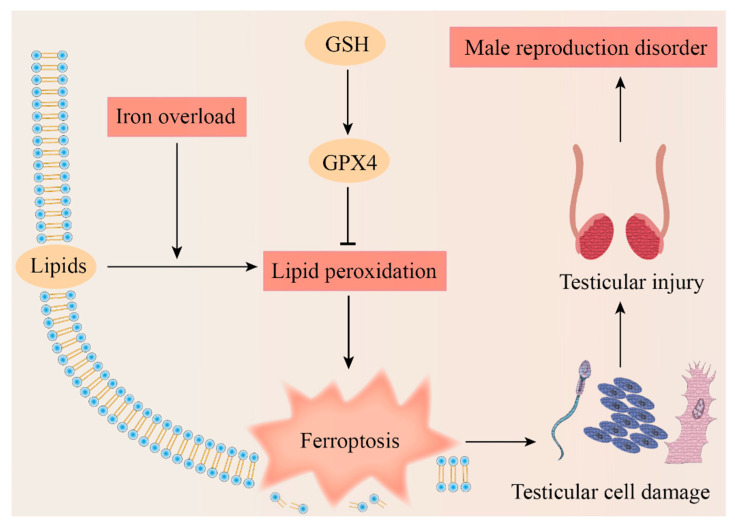
Potential mechanism of ferroptosis leading to male reproduction disorders. The figure illustrates that male reproductive disease could be caused by ferroptosis which results from iron overload, dysregulation of the Cyst(e)ine/GSH/GPX4 axis, and lipid peroxidation. GSH, glutathione; GPX4, glutathione peroxidase 4.

**Table 1 genes-14-00043-t001:** Morphological changes in testicular tissue induced by risk factors.

Factors	Experimental Subjects	Age	Dose Expose	Gross Changes	Microscopic Morphology Changes	Sub-Microscopic Changes	Ref.
Smoking	Human	Adults	Heavy smoke	N.A. *	Decreased sperm vitality and sperm progressive motility	N.A.	[6]
Busulfan	Mice	8 weeks old	4 mg/kg	Reduced testicular mass	A loss of spermatocytes and spermatids in the seminiferous tubules, increased lumen diameters and vacuolization	N.A.	[5]
PM 2.5	Mice	6–8 weeks old	431.4 μg/m^3^	N.A.	Sertoli cell vacuolization, immature germ cells, decreased number of Sertoli cells in the stage VII seminiferous tubules	N.A.	[7]
	TM4 Sertoli cells	—	100 μg/mL	—	Morphology of TM4 cells become agminated and irregular		
Cdmium (Cd)	Mice	From 10 to 27 weeks old	0.5 mg/kg	No changes	Slight detachment of germ cell	N.A.	[31]
			5 mg/kg	Lower testis weight	Partial detachment of germ cells, devoid of spermatogenic cells		
Arsenite	Mice	7 weeks old	50 mg/L	Declined testicular weight	Decreased sperm number, but no decrease in sperm malformation rate	Ruptured mitochondrial membrane, broken or even disappeared mitochondrial ridge	[83]
Di-(2-ethylhexyl) phthalate (DEHP)	Mice	3 weeks old	500 mg/kg	Decreased testes organ coefficient	Shrunken spermatogenic tubules and decreased Sertoli cell number	N.A.	[89]
Bisphenol A	Mice	8 weeks old	100 mg/kg and 200 mg/kg	Decreased testicular weight	Decreased sperm concentration and motility and increased deformity rate	Swollen mitochondria and increased density membrane	[60]

N.A.* indicates data not available.

## Data Availability

Not applicable.

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
