# Peer review of "Ferroptosis: A Novel Type of Cell Death in Male Reproduction"

_genes, 2022, doi:10.3390/genes14010043_

Round 1

Reviewer 1 Report

Ferroptosis reviewer report

My comments on this paper are mainly editorial; the science information itself generally seems fine.  However, in Table 1, the authors need to give more information as to how they know that there are ‘ferroptosis-related morphological characteristics’ (and therefore potentially a ferroptosis process occurring) for those agents where there is no available data for ‘gross changes’ or for ‘sub-microscopic changes’, and only non-specific ones in ‘microscopic morphology changes’. 

By their nature of collating information from the literature, reviews are often not easy for a non-specialist to read.  While this review seems to be comprehensive, unfortunately the reading of it is made doubly-difficult by the multitude of English language errors.  The authors must now, and in any future paper before submitting for review, ask a good English speaker to read through their manuscript and make appropriate language corrections.

In several places the meaning of a sentence is ambiguous and is occasionely even in danger of being interpreted opposite to the meaning intended.  In most cases this is due to missing words, or it not being clear from the text how additional clauses fit in.

I have picked out and listed in the Table below (and attached as a pdf) over 70 places where the language needs correcting. 

Line

Present text

Suggested revised text

Comment

29

hotpot

hotspot

31

 Once certain processes disorder presents

English - not sure what is meant here; needs rewriting

32

 barriers of glutathione

(?) barriers to glutathione

(?)

33

ROS

ROS (reactive oxygen species – which are…..)

Need to clarify this abbreviation here.

If it is ‘Reactive oxygen species’ – also indicate what these are.

34-35

 general process composed with various mechanism

General process arising from various mechanisms

37-38

 it has also been proved

This needs a reference

38-40

 Different factors induce ferroptosis resulting in morphological structure or functional dysfunction of reproductive cells, which is closely related to the occurrence of related diseases, such as male azoospermia and oligospermia

This needs a reference

57

BJ-derived

Define ‘BJ’ 

59

which induce a particular cell death

which induces a particular process of cell death

I presume the authors mean ’a particular type of process of cell death’  rather than ‘death of a particular cell type’

62-63

as classic ferroptosis agonists by

as a classic ferroptosis agonist by

Singular, not plural

68-69

the functions and mechanisms of ferroptosis continue to be deepened and updated.

our appreciation of the depth of the functions and mechanisms of ferroptosis continues to be updated.

75

in details

In detail

76

swearing

sweating  (??)

80

can be seen in ferroptotic cells compared to normal cells

Either:  …can be seen more in ferroptotic cells, compared to normal cells…

OR ‘…can be seen in ferroptotic cells, but not in normal cells’.

84

Nowadays,

New paragraph

85

with the company of rising diseases.

with the rising number of associated diseases

90

organ,

organs,

93

It is very promising

Potentially, It will be very promising 

97Table 1

N.A. for gross changes and morphology changes.

If the only changes are non-specific ones in Microscopic Morphology (as with ‘Smoking’), how do the authors know these are ferroptosis-related ?  

116 & 118

irons

ions

119

sixtransmembrane epithelial antigen

Do the authors mean:
 ? transmembrane epithelial antigen 6  ??

Please check and clarify

119

the prostate member 3 (STEAP3).

STEAP3 what ?  ... eg. STEAP3 gene ??  or 'metalloreductase enzyme ? 

Member ??

Please recheck what is the correct term in full here.

Are they referring to a protein or a gene ?

120

Following, Fe2+

Following this, Fe2+

128

clusters synthesis[

cluster synthesis

138 & 143

spermatocyte…..

spermatocytes

All should be plural

138 & 139 & 140

spermatid

spermatids

All should be Plural

140-141

Sertoli cells stores

store

Singular

142-143

could be supplemented, which is taken in by

Is ‘supplemented’ the correct word ? .  It could either mean ‘added to by something else’ (in which case the authors should say by what), or ‘added to by more iron’   The authors need to clarify this.

143

which is

Do the authors mean: ‘and is’ ??

153

Degradation of the only export FPN

What do the authors mean by :  ‘the only export ferroportin ?’  Is it the only export from the cell, or the only exporter carrying iron ?  Should 'export' be 'exporter' ?

164

lead to

leads to - present tense

led to - past tense

Which do the authors wish to use ?

168-169

sperm abnormalities including sperm vitality and progressive motility

including problems with sperm abnormalities, such as sperm vitality….

170-171

models, studying…

models, made for studying…  OR  models, which replicate or mimic….

180-181

mentioned in morphologic features (Table 1.)

mentioned in Table 1.

182

DFO

?? desferrioxamine (DFO)

DFO abbreviation needs indicating in full

186

TM4 cells, testicular Sertoli cells

TM4 cells, and in testicular Sertoli cells….

188

pm2.5[13].

pm2.5 [13]

Please also indicate what is pm2.5.

191

Round spermatids appeared iron-dependent cell death

Round spermatids showed (?? or ‘revealed’ ?,  or ‘appeared to show’) iron-dependent….

211

and inhibiting it such as using iron

and inhibiting it, such as by using iron……FPN, is likely…..

218

introduced

introduce

Present tense is better

219

summarized

summarize

Present tense is better

227

rate-limiting enzymes

rate-limiting enzyme

Singular

235-236

dipeptidase (DP) which towards the extracellular membrane[44].

dipeptidase (DP) which moves towards….

OR …migrates towards…

Verb is missing

236

reverted

recycled  ???

240

in male production

in male sperm (?) production (??)

248

straightly,

‘directly’  OR ‘immediately’ ? 

252

seminiferous due

seminiferous tubules ?

OR  ‘semen’

256

treat

treating  ? OR treatment ?

257

demonstrated the susceptibility

demonstrated that the susceptibility

264

taken from extracellular,

taken into cells from extracellularly

265-266

is proved limited[54].

Is proved to be limited [54].

275

could against

could protect against ??

Verb is missing

276

in later stage,

‘in later stages’,  OR ‘at a later stage’

300-301

In the mice model of oligozoospermia, which were given testicular injection of 300 busulfan,

a)In the mouse model of oligospermia, which is mice given testicular injection…

OR

b)In the mouse model of oligospermia, in those mice then given testicular injection…

Do the authors mean a)that the mouse model is created by the testicular injection ?
OR

b) Do they mean that the testicular injection was given to the already-existing mouse model ?  

This needs clarifying as suggested.  

307-308

when considering ferroptosis as a treatment for male infertility,

when considering inhibition of ferroptosis as a treatment….

do the authors mean ferroptosis; or do they mean inhibition of ferroptosis as the treatment ?

315-318

Phospholipid hydroperoxides… …disorders and ferroptosis

This sentence is too complex in structure, and very difficult to find its sense.  Please re-write.

320-323

When the capacity…will be provoked

This sentence is also too complex in structure, and very difficult to find its sense.  Please re-write.

323

It can occur by both non-enzymatic driven

This can occur by both non-enzymatic means (OR ‘processes’ ?) driven….

342

whose derivatives attach importance

whose derivatives are of importance

358

different those

different from those

362

structure altered, it

structure is altered, it…

363

negatively affects their

negatively affecting their

364

sperms are lack of related

sperms have a lack of related

OR

sperms are lacking related

375

shown that it exists PUFA incorporation into

shown the existence (OR occurrence ?) of PUFA incorporation into…

383

currently identified ferroptosis occurred in male

….ferroptosis occurring in the male…

402

by-product lipid peroxide

by-products of lipid….

408

thus resulting diseases in male

thus resulting in diseases of the male….

414-415

preventing lipid peroxidation process or running out of LIP as targets for treatment intervention point of ferroptosis, may be of potential value treating

preventing the lipid peroxidation process, or running…..targets for any treatment-intervention-point in ferroptosis, may be of potential value in treating….

425-426

Whether the different sensitivity to ferroptosis of  different cells during spermatogenesis is one of the

Is the different sensitivity… …spermatogenesis one of the….

428

Relative research

‘Relevant research’   OR

‘Related research’

430-431

There may exists a vast untapped terrain of ferroptosis which has a broad potentially prospect.

Unfortunately this sentence does not make any sense. Please re-write.

Reviewer 2 Report

The authors reviewed a novel type of cell death in male reproduction; however, the following questions should be clarified before decision:  

Minor revisions

1.      It would be good if you could write the type of current study (e.g. experimental study, etc.) in the both abstract and materials & methods sections.

2.      Write the abbreviation of all words where they are mentioned for the first time within the manuscript (abstract & text body).

Major revisions

1.      Figure 1 is not mechanistic. Please complete this figure using more mechanistic pathways.

2.      It would be well if you could show the process of papers’ selection, selection criteria by which you have chosen articles for the study.

3.      Creating tables summarizing experimental studies and results for each section is mandatory. In another word, you could make a table for each pathophysiology in which studies and the outcomes are summarized.

4.      Creating some figures to show mechanisms of each pathophysiology would be great.

5.      English editing of the manuscript is necessary.

Round 2

Reviewer 1 Report

Ferroptosis Re-review report Dec 2022
(this re-review report is also attached as a PDF file to ensure the original Word file formatting is retained)

The authors have attended satisfactorily to the corrections which I suggested in my original review, although some of the additional re-written passages do need further attention to the English, as indicated below. However, in noting also the comments of Reviewer 2 on the original version, I would agree that, as a literature review, the authors do need to explain in this paper, rather than just in their cover note, what their method of approach to researching the literature.

Points
1. The above would seem best achieved by adding a ‘Materials & Methods section’ between Section 1 (Introduction) and Section 2 (The discovery of ferroptosis).  Alternatively if the authors prefer to do this as Supplementary Material, but at the least, referred to by inserting a suitable heading between section 1 & 2, that would seem acceptable.

2. In addition to the overall list of abbreviations, each of the Figures would benefit from having some information as a Legend, not just the title of the Figure.  In particular this should include a guide to at least some of the abbreviations used in the Figure, particularly where these may be less familiar, as a Figure is much easier to follow where that information is immediately available attached to it.  

3. The overall list of abbreviations would work better for the reader if they were listed in alphabetical order (by abbreviation) rather than in order of first appearance in the text.   

4. Multiple Corrections to the English (or removal of ambiguity) are again required, particularly in the re-written text, but also some in the ‘auto-polished’ text.  These are (* The Line number refers to the supplied pdf version which includes the 3 different coloured types of mark-up) :

Line*

Present text

Suggested revised text

Comment

17

In this review

In this literature-based  review

To amplify reviewer 2’s comment re defining the type of study.

19

abnormal upon iron metabolism

Delete ‘upon’

146-148

It is worth noting…..

Delete one copy of this sentence

Sentence has been duplicated

161

Sertoli cells store the acquired iron in ferritin, which would be secreted and return to primary spermatocytes.

Should the word be 'returned': ie. that the iron is returned to the primary spermatocytes ?

This could currently read that the Sertoli cells would return to (being) primary spermatocytes, or return to be adjacent to primary spermatocytes.

204

F.Sun et al

?? Zhao et al

Is this reference 33, or is there a paper of Sun et al that needs citing as a separate reference ?

240

pf

of

269

ferroptosis in male productive system

ferroptosis in the male reproductive system

325

sperms

sperm

See comment below against lines 395-8

348

And the process…

The process…

Sentences cannot start with ‘And’.

395-8

Sperm has high content of….which are different from those…., so it is not….conclude that they are more vulnerable….

Sperm have a high content of…..

[then one of:]

which are different ones from those…., so it is not…conclude that the sperm are more vulnerable….

[OR:]
which are different ones from those…., so it is not…conclude that the PUFA are more vulnerable….

[OR:]

Sperm have a high content of…..

which is a different situation from somatic cells, so it is not…conclude that the sperm are more vulnerable….

The term ‘sperm’ (like the term ‘sheep’) covers both singular and plural.

Also, it is currently ambiguous whether the authors mean that having a high content of PUFA in sperm is a different situation from somatic cells; or whether the types of PUFA in sperm are different from those in somatic cells.

402-4

And

407

Moreover, sperm has a lack…

….it is unable…. 

…for their deficiencies

[Either:]  Moreover, a sperm has a lack… …it is unable… 
for a sperm’s  deficiencies

[OR:]
Moreover, sperm have a lack…  …they are unable… 
…for their deficiencies

See above

421

sperms

sperm

444

sperms

sperm

464

sperms

sperm

464

…to evaluate the male reproduction.

…to evaluate male reproduction.

465

…in male reproductive system..

…in the male reproductive system…

466

Here, we summarized…

Here, we have summarized…

467

…factors from...

…factors, classed as…

[OR]…factors, grouped as…

[rather than ‘factors from’]

468

In human,…

[Either:] In the human,…

[OR:] In humans, …

474

…in microscopic level…

…at microscopic level…

475

…broken or disappeared mitochondrial ridge…

[Either:]

'...a broken or lost mitochondrial ridge...' 
[OR:]

 '...a broken mitochondrial ridge, or its disappearance...'

477

Increased density membrane

Increased membrane density

[OR:]
Membrane of increased density

479

…other forms of death

…other forms of cell death

485

…can be almost detected

…can often be detected ??

What do the authors mean here by 'almost'? - either they are detected or they are not.

488

And SOD content is found…

Also, SOD content is found..

Sentences cannot start with ‘And’.

499-501

…the exposure of… 

 …can elicit mouse testes ferroptotic response through…

…exposure to…

[Either:] …can elicit the ferroptotic response in mouse testes through

[Or:]
…can elicit the same ferroptotic response as seen in mouse testes…

[Or:]
…exposure of mouse testes to…  …can elicit a ferroptotic response through….

It is ambiguous here whether the authors are still referring to human testes exposed to….  which show a response similar to ferroptosis in mouse testes;
..or whether they have jumped subject and are referring purely to mouse testes.

515

The question that we raise are…

The questions that we raise are…

538-581

Abbreviations

Please list in alphabetical order

Reviewer 2 Report

This manuscript has been revised accordingly. It is now acceptable. 
